# Bootstrap Sampling Improves Model Soup Performance via Increased Model Diversity for Pneumonia Classification

**Sara Early**[1,2,3]                                      SARA.EARLY@UCALGARY.CA
**Omata I. Ehizokhale**[4]                     IKHIANOSEN.EHIZOKHAL@UCALGARY.CA
**Nils D. Forkert**[2,3,5]                            NILS.FORKERT@UCALGARY.CA
**Samira Ebrahimi Kahou**[4,6]          SAMIRA.EBRAHIMIKAHOU@UCALGARY.CA

[1] *Biomedical Engineering Graduate Program, University of Calgary, Calgary, AB, Canada*

[2] *Department of Radiology, University of Calgary, Calgary, AB, Canada*

[3] *Hotchkiss Brain Institute, University of Calgary, Calgary, AB, Canada*

[4] *Department of Electrical and Software Engineering, Schulich School of Engineering, University of Calgary, Calgary, AB, Canada*

[5] *Department of Clinical Neurosciences, Cumming School of Medicine, University of Calgary, Calgary, AB, Canada*

[6] *Mila – Quebec Artificial Intelligence Institute*

## Abstract

Model soups combine multiple trained neural network checkpoints through weight averaging, often outperforming individual models and achieving performance comparable to deep ensembles without increasing inference cost. However, their effectiveness depends critically on checkpoint diversity, and when models are trained on the same dataset, optimization trajectories may converge toward similar regions of parameter space, limiting this diversity. In this work, we investigate bootstrap resampling as a simple data-level mechanism for increasing checkpoint diversity. Using a binary pneumonia classification task and 644 radiographs from the National Institutes of Health (NIH) ChestXray14 dataset, we train pools of convolutional neural networks under varying bootstrap ratios and construct greedy model soups. While checkpoint models trained on the full dataset achieve the highest mean individual accuracy, they are highly similar and offer little complementary signal, limiting the effectiveness of greedy selection. Bootstrap sampling introduces variability in the training data, producing more diverse checkpoints that, although individually weaker, enable greedy soup construction to combine complementary representations and achieve superior overall performance. The strongest model soup, obtained with 70% bootstrap sampling, achieves a test accuracy of 0.650, representing a 9.8 percentage point improvement over the mean individual checkpoint accuracy (0.551) under the same condition. While absolute performance is limited by the small cohort size and training-from-scratch setting, this result highlights the substantial gains achievable through diversity-driven weight averaging.

**Keywords:** Model soups, bootstrap sampling, ensemble learning, pneumonia classification

## 1. Introduction

Deep ensembles are known to improve predictive performance and calibration robustness compared to a single model, but their inference and training costs scale linearly with the number of ensemble members (Lakshminarayanan et al., 2017; Ovadia et al., 2019; Zhou, 2012). Model soups address this limitation by greedily selecting and averaging the weights

of multiple trained neural network checkpoints into a single model, preserving the inference cost of a single network while often recovering accuracy and robustness gains comparable to those of deep ensembles (Wortsman et al., 2022; Izmailov et al., 2018; Maddox et al., 2019). In greedy soup construction, checkpoints are ranked by validation performance and iteratively added to a running weight average only if they improve it.

However, the effectiveness of model soups depends critically on the diversity of the candidate checkpoints.(Kuncheva, 2003; Wen et al., 2020) Prior work in computer vision introduces diversity through hyperparameter variation, data augmentation, or fine-tuning schedules (Wortsman et al., 2022). In medical imaging, models trained from scratch on small datasets often converge to similar regions of the loss landscape (Goodfellow et al., 2016; Fort et al., 2019; Zech et al., 2018), limiting checkpoint diversity and the effectiveness of greedy soups. Bootstrap resampling may mitigate this by introducing variation in the empirical training distribution across runs, encouraging checkpoints to develop complementary representations and decision boundaries.

Bootstrap resampling has been used in classical ensemble methods to increase robustness by training models on resampled versions of the data (Breiman, 1996; Dietterich, 2000). However, its benefit for deep learning weight averaging on medical imaging datasets remains relatively unexplored. Rather than directly measuring checkpoint diversity in parameter space, we investigate whether bootstrap resampling improves greedy soup performance as a downstream indicator of increased effective diversity among candidate checkpoints.

We study this using chest radiographs from the National Institutes of Health (NIH) ChestXray14 dataset (Wang et al., 2017) by training model pools under varying bootstrap ratios and constructing greedy model soups. We compare soup performance against two baselines: the mean and best individual checkpoint model under each bootstrap condition, as well as checkpoints trained on the full dataset without resampling.

## 2. Methods

### 2.1. Dataset

Chest radiographs were obtained from the NIH ChestXray14 dataset (Wang et al., 2017). We constructed a balanced binary classification task using images labeled exclusively as *Pneumonia* or *No Finding*, resulting in 644 radiographs. Data were split into training, validation, and test sets using a stratified 70/15/15 partition. Images were downsampled to $224 \times 224$ and normalized.

### 2.2. Ensemble Training and Evaluation

All experiments use a 2D adaptation of the Simple Fully Convolutional Network (SFCN) (Peng et al., 2021), comprising five convolutional blocks with batch normalization, ReLU activations, and max-pooling, followed by a $1\times1$ convolution bottleneck, global average pooling, dropout, and a final classification layer. For each bootstrap ratio $\rho$, we first select a subset of $\rho N$ examples from the original $N$-sample training dataset without replacement. We then construct a bootstrap training set by sampling $N$ examples with replacement from this subset. We evaluate $\rho \in \{0, 0.5, 0.7, 0.9, 1.0\}$. The no-bootstrap condition ($\rho = 0$) trains each model on the full dataset without resampling, while $\rho = 1.0$ corresponds to classical

bootstrap sampling. For each bootstrap condition, we train a pool of $K = 72$ models, each with a distinct bootstrap sample of the training data, following prior model soup work (Wortsman et al., 2022) and to ensure a sufficiently large and diverse candidate set for greedy selection. For each model, hyperparameters are selected via a random search over 25 training trials, after which the best configuration is retrained to convergence. Training uses AdamW optimization with linear learning-rate warmup, step decay, and early stopping.

## 3. Results and Conclusion

Moderate bootstrap ratios (70–90%) improve greedy model soup performance as shown in Table 1. Considering mean individual accuracy, no-bootstrap training yields the strongest individual models (Acc 0.572), but the corresponding greedy soup (Acc 0.567) fails to improve over this mean. Bootstrap conditions produce individually weaker models on average, but yield substantially better greedy soups, suggesting that the diversity introduced by resampling enables greedy selection to identify a more complementary subset of checkpoints.

Table 1: Greedy soup and individual model performance across bootstrap conditions ($K = 72$). $\Delta$Acc is greedy soup accuracy minus mean individual accuracy.

| Condition | Mean Acc | Mean AUC | Best Indiv Acc | Greedy Soup Acc | Greedy Soup AUC | $\Delta$Acc |
|---|---|---|---|---|---|---|
| No bootstrap | $\mathbf{0.572} \pm 0.053$ | $\mathbf{0.597} \pm 0.058$ | 0.649 | 0.567 | 0.617 | $-0.005$ |
| Bootstrap 50% | $0.562 \pm 0.060$ | $0.573 \pm 0.066$ | 0.670 | 0.567 | 0.592 | $+0.005$ |
| Bootstrap 70% | $0.551 \pm 0.062$ | $0.557 \pm 0.069$ | $\mathbf{0.680}$ | $\mathbf{0.650}$ | 0.625 | $+\mathbf{0.098}$ |
| Bootstrap 90% | $0.554 \pm 0.055$ | $0.571 \pm 0.060$ | 0.649 | 0.608 | $\mathbf{0.657}$ | $+0.054$ |
| Bootstrap 100% | $0.566 \pm 0.053$ | $0.592 \pm 0.058$ | $\mathbf{0.680}$ | 0.546 | 0.620 | $-0.020$ |

We find that $\rho = 0.70$ and $\rho = 0.90$ substantially improve greedy soup performance. At $\rho = 0.70$, the greedy soup achieves a test accuracy of 0.650 and ROC-AUC of 0.625, improving by 9.8 percentage points over the mean individual checkpoint accuracy of 0.551 under the same bootstrap condition. At $\rho = 0.90$, the soup achieves a test accuracy of 0.608 and the highest ROC-AUC overall (0.657). Both extremes underperform: at $\rho = 0$, the candidate pool is too homogeneous for greedy selection to identify a meaningfully distinct subset, while at $\rho = 1.0$ individual model quality degrades sufficiently to offset any diversity benefit. Overall, our results suggest that bootstrap resampling can improve greedy model soup performance when training from scratch on small medical imaging datasets, with moderate ratios balancing candidate diversity against individual model quality. The observed improvements are consistent with a diversity-based explanation, though we did not directly quantify this in parameter space. Experiments on a single dataset limit generalizability, and future work should examine larger datasets, other tasks, such as segmentation or multi-label classification, and the combination of bootstrap resampling with pretrained initializations.

## Acknowledgments

We thank NSERC, CIHR, Alberta Innovates and DRAC for providing funding and compute.

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
