# OpenReview forum: "Bootstrap Sampling Improves Model Soup Performance via Increased Model Diversity for Pneumonia Classification"
_MIDL.io/2026/Short_Papers — MIDL 2026 - Short Papers Poster_

### Official Review · Reviewer_juEY · 2026-04-23
**Simple, well-executed**

**Rating:** 5
**Confidence:** 4

**Review:**

Quality/Clarity: The proposed method is presented clearly. Using a pool of 72 models per bootstrap ratio and a thorough hyperparameter search provided a reasonably robust experimental basis.

Significance/Originality: The evaluation is currently limited to a single dataset and would benefit from additional ablations and comparison to the existing literature. That said, the results on this dataset appear robust, and the approach has potential applicability to other medical imaging scenarios that rely on training from scratch.

**Summary:**

Model soups average the weights of multiple trained models to produce a potentially stronger combined model. As with ensembles, the effectiveness of the resulting model depends on the diversity of the constituent checkpoints. The authors propose using bootstrap resampling, selecting $\rho$% of the training data to train each model from scratch, and combining the resulting checkpoints via model soups. Using a pool of 72 models, they demonstrate that bootstrap resampling improves model soup performance over training on the full dataset. On the NIH ChestXray14 dataset (644 radiographs), the reported gains are approximately 8 percentage points in accuracy and 4 percentage points in AUC.

**Strengths:**

- The idea is simple, well-motivated, and clearly presented.
- Performance gains are demonstrated across multiple bootstrapping ratios, supporting the generality of the finding.
- The experimental design, including hyperparameter tuning, appears careful and thorough.

**Weaknesses:**

- No comparison is made against other model soup approaches, and potential interactions with existing methods are not discussed. It would be interesting to know whether bootstrap resampling could further improve other soup variants.
- It is unclear whether the observed benefits would extend to settings where pretrained models are used rather than models trained from scratch.

**Justification Of Rating:**

The paper presents a simple and clearly motivated idea, supported by solid experimental results. Despite the current limitation to a single dataset, the contribution is promising and well-executed.

---

### Decision · Program_Chairs · 2026-05-08

Accept (Poster)